5

# Agricultural uses reshape soil C, N, and P stoichiometry in subtropical ecosystems

H. Y. Liu<sup>1,\*</sup>, J. G. Zhou<sup>1,\*</sup>, J. Shen<sup>1</sup>, Y. Y. Li<sup>1</sup>, Y. Li<sup>1</sup>, T. D. Ge<sup>1</sup>, G. Guggenberger<sup>1, 2</sup>, J. Wu<sup>1</sup>

<sup>1</sup>Key Laboratory of Agro-ecological Processes in Subtropical Region, Institute of Subtropical Agriculture, Chinese Academy of Sciences, Hunan, 410125, China

<sup>2</sup>Institute of Soil Science, Leibniz Universit ä Hannover, 30419, Hannover, Germany

\*These two authors contributed equally to this work.

Correspondence to: J. Wu (jswu@isa.ac.cn; mailto:gtd@isa.ac.cn)

Abstract. Changes in elemental stoichiometry, in most cases, attributed to land use alterations may cause vital impacts on 10 the nutrient status and environmental quality of ecosystems. Here, we studied the stoichiometry and spatial distribution patterns of soil organic carbon (SOC), total soil nitrogen (TN), and total soil phosphorus (TP) in topsoil (0-20 cm; 1207 samples) ecosystems in a representative catchment of subtropical hilly region of China. Its main land uses are woodland, paddy fields, and tea farmlands. Data obtained show that the medians of SOC, TN, and TP were 16.97, 1.83, and 0.52 g kg<sup>-1</sup>, and medians of C:N, C:P, and N:P molar ratios were 10.0, 78.6 and 7.9, respectively. The best-fitting model were exponential 15 models for SOC, TN, TP, C:N, and N:P, while for C:P was Gaussian model. The nugget values for SOC, TN, TP, C:N, C:P, and N:P were 1.0, 0.06, 0.01, 6.0, 56.0, and 1.0, respectively. And their ranges were 750, 1290, 570, 2970, 810, and 720, respectively. The nugget-to-sill ratio (NSR) for SOC, TN, TP, C:P, and N:P were 2.7%, 14.3%, 20.0%, 4.0%, and 10.0%, respectively, and showed strong spatial autocorrelation. While C:N molar ratios had a moderate spatial correlation, with NSR of 49.95%. Spatial analyses showed that agriculture derived land use changes alter largely the spatial distribution and 20 stoichiometry of C, N, and P elements in individual landscapes and entire catchment. For woodland ecosystems, topography factors (elevation and slope) determined the elemental spatial distributions and stoichiometry (C:N, C:P, and N:P molar ratios). However, this status had been merged in agricultural ecosystems, due to the relative similarity in cropping and managing (N and P inputs through fertilization). Agriculture significantly increases N, and P contents but narrows C:N, C:P, and N:P molar ratios. Thus, our findings demonstrate that agricultural activities can affect carbon and nutrient stoichiometry 25 at the catchment scale.

# 1 Introduction

The cycling of C, N, and P and their interactions in ecosystems play vital roles to maintain the primary ecological

development of relevant and reliable information.

5

productivity and also influence the environment status and global climate changes (Thornton et al., 2009; Griffiths et al., 2012), particularly through N and P transfers from terrestrial sources and sinks to water bodies; and through C sequestration or the emissions of green-house warming gases including CO<sub>2</sub>, CH<sub>4</sub>, and N<sub>2</sub>O (Freeman et al., 1992; Cameron et al., 2002; Matamala et al., 2008). The ecological stoichiometry involving these elements provides an integrative nutrient framework for linking biogeochemical patterns in ecosystems (Sterner and Elser, 2002), and its changes can greatly affect the nutritional and environmental status of soil. It is considered to be very valuable for understanding the stability of terrestrial ecosystems, as for individual ecological types (e.g., grasslands and forests) and the global scale (e.g., Elser et al., 2000; McGroddy et al., 2004; Cleveland and Liptzin, 2007).

The C, N, and P contents in ecosystems including in soil and vegetation compartments are generally related (Vitousek,
2004), and in return, can be affected by ecosystem processes. Inputs and patterns of C, N, and P are the primary processes that determine the stoichiometry of these elements in ecosystems (Sardans et al., 2012). For natural ecosystems, C and N inputs are derived from the atmosphere via photosynthesis, N fixation, and dry and wet deposition, while P input are derived mainly via the weathering of soil minerals (Lajtha and Schlesinger, 1988; Li et al., 2012). These processes are largely dependent on geo-climatic regimes, which regulate water availability, soil parent materials, and plant photosynthetic capacity.
The status of nutrient stoichiometry along ecosystem chronosequences can be strongly modified by both natural and anthropogenic disturbances. A good example for this involves observations from Hawaii where the results demonstrated that the C:N:P ratio in the soil of a relatively young ecosystem was controlled mainly by the N supply, while that of a relatively

Land uses also affects, to a large extent, the C, N, and P inputs and consequently the elemental stoichiometry in soil,
particularly for farmed lands, where crop types and fertilization practices change frequently along with the biogeochemical processes in soil (Rezaei and Gilkes, 2005; Griffiths et al., 2012). Such variations can happen at a small geographical scale such as in small water catchments, based on a study reported by Li et al. (2013). Small catchments represent well-defined landscape units with respect to their boundaries, and they are ideal for conducting insightful investigations into the patterns and soil C, N, and P status and the associated stoichiometry (Corstanje et al., 2008). Such work can clearly reflect the impacts on the environment (e.g. the extents of N and P richness) and global climate change (through C sequestration) from agriculture systems, both aspects appear increasingly as great concerns worldwide and require scientific attention for the

old ecosystem was governed mainly by the P supply (Herbert and Fownes, 1995; Vitousek and Farrington, 1997).

30

The subtropical region in China hosts the world most intensive agricultural systems, including mainly double or even triple crop rotations in a year, such as double rice, single or double rice plus oil-seed rape, rice plus wheat or maize in paddy fields, 4–6 vegetables rotations, and well-managed orange and tea farmlands. All these agricultural systems have received high inputs of chemical fertilizers (usually over 500 kg N and P ha<sup>-1</sup>) during the last three or four decades, and heavy surface pollutions is apparent over much of the region (MEPC, 2010). This region has relatively high precipitation (usually

5

10

1200–1450 mm yr<sup>-1</sup>; Liu et al., 2012) and consequently developed in complex aquatic networks which enhance N and P transferences and respond to varying degrees to aquatic pollution. Small catchments are the most essential topographic unit in hilly lands (which account for over 60% of the total area) of the subtropical region in China, and they are particularly relevant as a basic land unit in scientific studies. Specifically, data from small catchments can reflect the ecological status of soil C, N, and P stoichiometry and the variability caused by the influences of topographic conditions and anthropogenic disturbances via alternating land uses and inputs of N and P from cropping practices and soil fertilization. Our study was conducted in a representative catchment of the subtropical region of China, and the objective was to understand the extent to which intensive agriculture can change the soil C, N, and P stoichiometry, which can be useful for accessing the potential that agricultural practices can alter the ecological framework of C, N, and P stoichiometry, and environmental status in the subtropical region. This may also provide a valuable warming to balance agriculture with ecological and environmental conservation for other region worldwide.

### 2 Material and methods

#### 2.1 The catchment investigated, soil sampling and analyses

- The catchment studied in this work is located in Jinjing Town, Changsha County, Hunan Province, China, and covers an area of 134 km<sup>2</sup>. It has a subtropical monsoon humid climate with an average annual precipitation of 1300 mm and a mean annual air temperature of 16.9 °C. The catchment has a typical hilly topography with an elevation ranging from 66 m to 440 m. Land use mainly consists of woodlands (WL), paddy fields (PF), and tea farmlands (TF), which account for 58.5%, 31.6%, and 4.3% of the total catchment area, respectively. Other minor land uses in the catchment include water bodies, roads, and residential areas, which together occupy only 5.6% of the total catchment area.
- The random sampling design were established based on distributions of different topographic factors (elevation, slope, and TWI) and land use patterns at the whole catchment. There were low sampling density in the regions with a small variability of landscape composition and structure, and high sampling density in the regions with a large variability of landscape composition and structure. Generally, a number of 1207 samples of topsoil (0–20 cm) in the catchment were mostly collected by approximate the spacing distance (50–100 m) between sites during 2010 and 2011. Of these samples, 624 were from paddy fields, 515 were from woodlands, and 68 were from tea farmlands. For each sampling site, 5–8 soil cores within a 5 m radius were taken using a stainless steel anger (Ø 3cm), homogenized by hand mixing, and then sieved through a 2 mm sieve after air-dried at room temperature. The geographic position (longitude and latitude) of each sites were measured with a global position system (GPS, Sand-ing Southern Survey Co., China) meter and information on the agricultural land use and management practices around the sampling site's center were also recorded. The soil samples were analyzed for the contents
- 30 of soil organic carbon (SOC) and total nitrogen (TN) using a C/N elemental analyzer (Vario MAX, Elementar, Hanau,

Germany), and total phosphorus (TP) by the HClO<sub>4</sub> digestion molybdate-blue method (Lu, 2000).

# 2.2 Topographic and land use variables

5

The most recent digital elevation map (DEM, 5 m spatial resolution), obtained from the Land Surveying Department of the Hunan Province, was used to generate the elevation (Fig. 1a) and slope (Fig. 1b) variables for the study catchment. Land use map (also 5 m spatial resolution; Fig. 1c) for the study area was also obtained from the same source as DEM. For converting land use types into an explanatory variable as required by the interpolations of SOC, TN, TP contents, and their molar ratios (i.e. C:N, C:P, and N:P) in the catchment investigated, the presence or absence of three major types of WL, PF, and TF was represented by the dummy value of 1 and 0, respectively. The position of all sampling sites in the catchment was allocated as Fig. 1d.

# 10 2.3 Spatial analyses and spatial interpolations

Semivariogram model (Webster and Oliver, 2001) was used to analyze the spatial variations of the observed soil properties (i.e. SOC, TN, and TP contents) and their molar ratios (C:N, C:P, and N:P) for all sampling sites in the catchment studied. The semivariogram for each of the properties at all the sites was calculated using Eq. (1) as followings function:

$$\gamma(\mathbf{h}) = \frac{1}{2N(\mathbf{h})} \sum_{i=1}^{N(\mathbf{h})} [\mathbf{Z}(\mathbf{X}_i) - \mathbf{Z}(\mathbf{X}_i + \mathbf{h})]^2$$
(1)

where, N(h) represents the number for the pairs of the adjacent sampling sites ( $X_i$  and  $X_i + h$ ) separated by a distance (h), and  $Z(X_i)$  and  $Z(X_i+h)$  the values for each property at the sites  $X_i$  and  $X_i+h$ .

The nugget (N), sill (S), and nugget-to-sill ratio (NSR) are usually used to reflect the random and inherent errors, the total variability and the degree in the spatial dependence of variables required for geographical analysis (Webster and Oliver, 2001). In our work, N, S, and NSR for SOC, TN, TP, C:N, C:P, and N:P were derived from their best fitting model (i.e. linear, spherical, exponential, Gaussian, and rational quadratic models) for the plots of their semivariogram via spatial range, using the GS+ Version 9 software (Gamma Design Software, Plainwell, MI).

The spatial interpolations of SOC, TN, TP, C:N, C:P, and N:P in the catchment investigated were estimated geographically weighted regression (GWR) model developed by Fotheringham et al. (1998). Leave-one-out cross validation method was used to analyze the optical estimated values and their mean absolute error (MAE) and root mean square error (RMSE), while

Pearson correlation coefficient (r) used to test the prediction accuracy between the estimated values and observed values.

### 2.4 Classification and regression tree

To eliminate the collinearity effects in elevation and slope when analyzed the driving force variables, the new topography variable "ES" was first generated by merging them with the principal component analysis (PCA) method implemented with CANOCO 5.0 software (Microcomputer Power, Ithaca, USA). Importance indices of the influences of ES and land uses on C:N, C:P, and N:P molar ratios, and their regression tree for individual land use types (WL, PF and TF) were analyzed using classification and regression tree (CART) model as described in Therneau et al. (2010) and Breiman (2001; http://www.r-project.org).

### 2.5 Statistical analyses

One-way analysis of variance (ANOVA), Duncan's Multiple Range test, and Spearman's rank correlations of SOC, TN, and 10 TP, and their molar ratios were processed with SPSS 18.0.

# 3 Results

#### 3.1 Spatial distribution of SOC, TN and TP in the catchment

15

5

With different theoretical models (including linear, spherical, exponential, Gaussian and rational quadratic models) tested, the best-fitting models were exponential model for SOC, TN, TP. Their N values were 0.01, 0.06, and 1.0, respectively. The NSR for SOC, TN, and TP showed strong spatial autocorrelation (Cambardella et al., 1994). SOC exhibited the strong spatial dependence (NSR = 2.70%), with a range of 750 m, while TN had weaker spatial dependence (NSR = 14.28%), with a range of 1290 m. The weakest spatial dependence of 20.00% was found with TP, which had a shortest range (570 m) among the three parameters (Table 1).

20

The high values of SOC and low values of TN and TP were mainly distributed at relatively higher elevations and steeper slopes in the northern parts of the catchment where woodlands were the dominant land use type. The low values of SOC and high values of TN and TP were mainly distributed in the southern parts of the catchment where mainly distributes paddy fields and tea farmlands with low elevations and slow slopes (Fig. 2).

Across the catchment investigated in our work, the contents of SOC, TN, and TP ranged from 6.36 to 60.65, 0.66 to 3.90, and 0.18 to 1.87 g kg<sup>-1</sup>, and the corresponding medians as 16.97, 1.83, and 0.52 g kg<sup>-1</sup>, respectively. The coefficient variation

25 (CV) values, was calculated by Duncan's multiple range test, showed the variation of individual calculation for each contents. The CV values were relatively high, as 37% for SOC, 34% for TN, and 40% for TP (Fig. 3).

# 3.2 Spatial distribution of soil C:N, C:P, and N:P molar ratios

For C:N, and N:P the best-fitting model were exponential, while for C:P was Gaussian model, presumably due to the larger CV (22.4 to 250.6; Table 1). Their N were 6.0, 56.0, and 1.0, respectively. Geo-analysis using semi-variogram model shows that C:N molar ratios had a relatively large range, with a distance of 2,970 m and a moderate spatial correlation, with NSR of 49.95%. Compared to that, C:P and N:P had a smaller range of 810 m and 720 m, respectively, and strong spatial correlation with NSR of 4.03% for C:P and 10.00% for N:P(Table 1).

As indicated in Fig. 4, high values of soil C:N, C:P, and N:P molar ratios occurred in the woodlands with higher elevation and steeper slope, while low values of soil C:N, C:P, and N:P molar ratios occurred in agricultural uses of paddy fields and tea farmlands with low elevation and slow slope.

10

Soil C:N molar ratios ranged from 5.4 to 27.2, with a median of 10.0 and a CV of 31%. The C:P and N:P molar ratios ranged from 22.4 to 250.6 and 2.6 to 23.8, and had the medians of 78.6 and 7.9, respectively. Large CV values were found for both C:P (44%) and N:P (39%; Fig. 5).

# 3.3 Influences of topography and land use on soil C:N, C:P, and N:P molar ratios

Soil C:N, C:P, and N:P molar ratios were divided into 10 groups to ensure consistent numbers of soil samples for each group according to the distribution histograms of elevation and slope inclination. As shown in Fig. 6, mean values of soil C:N and C:P molar ratios were significantly lower in areas at lower elevations (< 115.9 m) than in areas at relatively higher elevations (> 115.9 m), and highest mean value of C:N molar ratios were observed at elevation of 180.8-416.1 m (p < 0.05). Mean values of soil N:P molar ratios also had no significant differences in areas at lower elevation (< 142.2 m; p > 0.05), and increasingly larger values in areas at higher elevations (> 142.2 m; p < 0.05).</p>

- 20 Mean values of soil C:N and N:P molar ratios were significantly higher for slopes exceeding an inclination of 23.4 ° than those for flatter slopes (< 23.4 °, p < 0.05). Effects of slope on soil C:P molar ratios also showed no significant differences among five classes with relative flatter slopes (< 8.3 °, p > 0.05), but increased with increasing steepness of the slope for slopes greater than 8.3 ° (p 

5

20

25

significantly (p 

only moderate spatial correlation.

In this study, we used the GWR method to make optimal, unbiased estimates of SOC, TN, TP, and their molar ratios at non-sampled locations. The Pearson correlation coefficients of all variables varied from 0.46 to 0.72, their MAE varied from 0.11 to 25.64, and the RMSE varied from 0.14 to 36.62 (Table 2). Overall, the correlation between the observed and predicted values was high, the model errors were small, and spatial interpolation results were basically similar to the actual situation.

- The soil C:N, C:P, and N:P molar ratios (10.0, 78.6, 7.9) in this study were similar to the results (10.0, 80.0, and 7.9) obtained by Li et al. (2012), and the results (12.1, 78, and 6.4) reported in Tian et al, (2010) for soils in the tropical and subtropical climatic zones of China. However, our values were much lower than soil C:N, C:P, and N:P ratios of 16.4, 286.5, and 17.5 reported in Xu et al. (2013), for 0–30 cm top soils at the global scale, that of 12.05, 219.0, and 18.13 reported in Griffiths et al. (2012) in a grazed grassland, and the result of 14.3, 186, and 13.1 reported in Cleveland and Liptzin (2007) for 0–10 cm mineral soils at the global scale. This was most likely attributed to the richness of soil N and P, as indicated by the fact that the median soil TN (1.83 g kg<sup>-1</sup>) and TP contents (0.52 g kg<sup>-1</sup>) in the catchment investigated in our work were quite large, compared with those reported for the global scale.
- 15

10

- Our results showed the mean soil C:N, C:P, and N:P molar ratios under agricultural uses (paddy field and tea field) were all significantly lower than the corresponding values for the woodland. Under natural conditions along with only minor human disturbances, SOC accumulated gradually due to the long-term leaf litter and root litter accumulation (Li et al., 2012), and thus the SOC contents were usually higher in the natural ecosystems than those in the agricultural ecosystems (Gao et al., 2013; Wang et al., 2014; Zhu et al., 2014). However, with no additional P input and relatively low amount N input from
- 20 atmospheric deposition (Shen et al., 2013) and biological N fixation, the soil TN and TP contents in the natural ecosystems are relatively low, especially for P (Gao et al., 2013; Wang et al., 2014). Therefore, the soil C:N, C:P, and N:P molar ratios are relative high. When the natural ecosystems are transformed to agricultural ecosystems (e.g. from woodland to tea farmland, from wetland to paddy field), SOC contents decreased due to increased soil organic matter mineralization as well as soil erosion caused by tillage (Zhu et al., 2014). In the agricultural fields in subtropical region, high application rates of N
- and P fertilizers are used for promoting crop yields. For example, the N and P application rates were 360 and 100 kg N ha<sup>-1</sup> yr<sup>-1</sup> respectively, for the paddy fields and 450 and 120 kg N ha<sup>-1</sup> yr<sup>-1</sup> respectively, for the tea fields. The long-term applications of N and P fertilizers have been proved to increase soil TN and TP contents in a large percentage (Griffiths et al., 2012; Qin et al., 2010; Tong et al., 2009), especially for P, as P is less mobile than N in the soil. Consequently, soil C:N, C:P, and N:P molar ratios are relative low in the agricultural land uses.
- 30

The increase in soil C:N, C:P, and N:P molar ratios with the elevation and slope raising at the catchment investigated (Fig. 6), suggests that elevation and slope may have positive effects on soil C:N, C:P, and N:P molar ratios. These finding are consistent with the results for a small watershed in northern China where both SOC and TN had significant positive

ratios in woodlands, but not in the paddy fields and tea farmlands.

correlations with elevation and slope (Gao et al., 2013). The phenomenon probably due to topography has a great impact on soil hydrology, soil physical properties, and chemical characteristics (Mulla, 1993; Murphy et al., 2011; Zhu et al., 2014), and it can strongly influence precipitation infiltration, the distribution of vegetation covers, and soil nutrients concentrations (Norton et al., 2003).

- 5 Though both land use and topography can affect the soil nutrients stoichiometry, our results showed that land use always had a higher importance value than topography of elevation and slope (Fig. 7), which indicates that it was the determining factor for soil C:N, C:P, and N:P molar ratios at the catchment scale. In our study area, woodlands usually occurred at higher elevations and on steeper slopes than the paddy fields and tea farmlands, which were located in the plains or in areas with low hills (Fig. 6). It was identified that for ecosystems under native vegetation (i.e., woodland), topography is one of the prime factors determining the spatial distribution and the elemental stoichiometry (C:N, C:P, and N:P molar ratios; Zhang et al., 2011; Hook and Burke, 2000). However, in the agricultural ecosystems, their effects have been submerged (statistically insignificant), due to the relative similarity in cropping and managing (N and P inputs through fertilization), which significantly increases soil N and P contents but narrows C:N, C:P, and N:P molar ratios (Gao et al., 2013; Griffiths et al., 2012; Wang et al., 2014). This helps to explain why topography had a positive effect on the soil C:N, C:P, and N:P molar
- 15

In the subtropical region, the typical land use types in a catchment scale mainly include woodland, paddy field, vegetable field, tea farmland or fruit garden. As shown in this study, agricultural uses had caused a lower and narrower soil C:N, C:P, and N:P molar ratios as compared to those values in the woodland. Therefore, agricultural uses reshaped soil C, N, and P stoichiometry in the subtropical ecosystems. Under the agricultural land use, a low soil C:N and C:P molar ratios usually means that a lower soil organic carbon content and higher soil TN and TP contents as compared to the woodland land use (Gao et al., 2013; Griffiths et al., 2012; Wang et al., 2014). Thus, in the subtropical region, transformation of woodland to agricultural lands should be discouraged to reduce soil carbon loss and also to decrease the risk of soil N and P loss by leaching and erosion. Besides, considering the high soil TN and TP contents in agricultural lands in the catchment, excessive N and P application rates should be avoided and also the soil residual N and P should be utilized to reduce the N and P accumulation (Chen et al., 2014; Sattari et al., 2012).

25

20

# 5 Conclusions

Elemental stoichiometry is a key indicator for the stability and divers status of the ecosystems. Topography (elevation and slope) and land use can affect the elemental stoichiometry (soil C:N, C:P, and N:P molar ratios) of soils in subtropical hilly catchments. We studied the spatial distribution of soil C, N, and P (carbon, nitrogen, and phosphorus) stoichiometry of the ecosystems in a subtropical catchment, and found that the stoichiometry was shaped in relatively narrow ranges in

5

agricultural uses, and its spatial variations with topography were remarkably reduced. Thus, our findings demonstrate that intensive agriculture can change the spatial distributions of soil C, N, and P and the associated stoichiometry in a hilly subtropical catchment.

*Acknowledgements*. This study was supported financially by the National Natural Science Foundation of China (41201299; 41430860), the Strategic Priority Research Program of the Chinese Academy of Sciences (XDB15020401), and the Recruitment Program for High-end Foreign Experts of the State Administration of Foreign Experts Affairs awarded to Prof. G. Guggenberger (GDT20154300073).

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
