# Peer review of "Agricultural uses reshape soil C, N, and P stoichiometry in subtropical ecosystems"

_Biogeosciences, 2016_

## Referee Comment (RC1) · T. Domingues (Referee) · 14 Jul 2016

The manuscript shows that in agricultural areas, due to inputs of fertilizers, soil nitrogen and phosphorus contents are higher than at natural woodlands. The authors go further and attempt to include landscape features (slope) as a explanatory variable for spatial variability in soil characteristics. This last objective was, in my view not fully explored as most of the natural woodland areas occur at high slope regions, while most of the agricultural areas occurs at low slope areas. I believe that the manuscript does not bring enough novelty to justify publication, although it presents a new and large dataset.

---

## Short Comment (SC1) · 15 Jul 2016

Based on extensive data, this study addressed influence of agricultural activities and topography on soil C, N and P stoichiometry in a subtropical hilly catchment in China using geostatistical method. Results indicated that agricultural activities shaped d narrower soil C:N, C:P, and N:P molar ratios in relatively narrow ranges and its spatial variations with topography were remarkably reduced. For woodland ecosystems, the spatial distributions of nutrient levels as well as stoichiometry were influenced by topography factors (elevation and slope). These results improve our understanding of influence of agricultural activities on soil nutrient dynamics, especially for stoichiometry. To enhance concerns, it is necessary to extend the discussion concerning implications of changes in nutrient stoichiometry from agronomic and environmental perspectives.

[Figure]

Remove "water" in "small water catchments" What means of the new topography variable "ES"?

---

## Short Comment (SC2) · 19 Jul 2016

This manuscript analyzed the spatial variation of soil C, N, and P stoichiometry in the subtropical watershed of Jinjing, China, and pointed out that the influence of agricultural land use on the spatial distribution and stoichiometry of soil C, N, and P at a watershed level. Considering the results derived from a subtropical watershed, it can't be extended to the whole of the subtropical region. So, the title of the paper should be limited in the certain subtropical watershed.
* * *

---

## Referee Comment (RC2) · S. Rolinski (Referee) · 21 Jul 2016

The manuscript aims at deciphering the impact of agricultural landuse and orography on the stochiometric ratios between C, N and P. The study presents an impressive dataset of soil samples in a catchment in a substropical hilly region in China. Although the methods are valid and statistical analysis sound, the conclusions are not convincing because of the distribution of woodland and agricultural areas. Surely, the long-term application of fertilizers is expected to have an impact on soil nutrient status but on the other hand, plantations for tea and rice were established primarily in the lowlands exactly because of more fertile soils. Apart from this dilemma, the presentation lacks clarity especially in the description of the methods.

---

## Author Comment (AC1) · 12 Sep 2016

We listed the requests from D.Chen and give the following responses.

Request 1:To enhance concerns, it is necessary to extend the discussion concerning implications of changes in nutrient stoichiometry from agronomic and environmental perspectives Response 1: In our opinion, we addressed these implications reasonably. Below, some discussion given in the manuscript is cited: P.2, Lines of 16-24 "A good example for this involves observations from Hawaii where the results demonstrated that the C:N:P ratio in the soil of a relatively young ecosystem was controlled mainly by the N supply, while that of a relatively old ecosystem was governed mainly by the P supply (Herbert and Fownes, 1995; Vitousek and Farrington, 1997)." P. 9, Lines of 20-27: "It was identified that for ecosystems under native vegetation (i.e., woodland), topography is one of the prime factors determining the spatial distribution and the elemental stoichiometry (C:N, C:P, and N:P molar ratios; Zhang et al., 2011; Hook and Burke, 2000). However, in the agricultural ecosystems, their effects have been submerged (statistically insignificant), due to the relative similarity in cropping and managing (N and P inputs through fertilization), which significantly increases soil N and P contents but narrows C:N, C:P, and N:P molar ratios (Gao et al., 2013; Griffiths et al., 2012; Wang et al., 2014)."

We did not go further with our discussion, as we would not like to be too speculative.

Request 2: Remove "water" in "small water catchments" Response 1: We have revised the manuscript accordingly.

Request 3:What means of the new topography variable "ES"? Response 3: The variable ES was generated by merging the two variables of elevation and slope with the principal component analysis (PCA) method implemented with CANOCO 5.0 software (Microcomputer Power, Ithaca, USA). We used the variable ES to eliminate the collinearity effects of elevation and slope, when analyzing the driving variables of the contents of soil C, N, P and their stoichiometry.

---

## Author Comment (AC2) · 12 Sep 2016

We repeat the request from X. Xu and give the following responses.

Request 1:The title of the paper should be limited in the certain subtropical watershed. Response 1: It is a good suggestion. We changed the title into "Agricultural land use reshapes soil C, N, and P stoichiometry: evidence from the subtropical Jinjing catchment , China"

---

## Author Comment (AC3) · 12 Sep 2016

Thanks to both reviewers for raising this important issue. In their both opinion, the conclusion about "agricultural uses reshape soil C, N, and P stoichiometry" is not convincing due to the distribution of woodlands in the uplands and tea and rice in the lowlands. The question is due to the fact that we did not explain the distributions of both the contents of soil C, N and P, and the area percentages of land use across the gradients of elevation and slope in detail in the last manuscript. Here, we will further explain "why agricultural uses reshape the contents of soil C, N, P and their stoichiometry" using more new Supplement materials. We also gave more presentations of used methods in the manuscript according to S. Rolinski's comments.

In the following we would like to argue, why in our opinion it is true that agricultural

land uses reshape the contents of soil C, N, P and their stoichiometry in the Jinjing catchment in a subtropical hilly region, China.

In subtropical, hilly regions of China, farming activities are from lowlands to high lands due to high population load and limited arable land availability. The same condition occurs in our research area, a classically subtropical, hilly catchment. In our research area, elevation varied from 59 to 416m and the slope from 0 to 71.6°. The majority of the region was below 200 m elevation and at a slope <30.6°. We calculated the area percent for paddy field (PF), tea farmlands (TF), and woodland (WL) under different gradients of elevation and slope. We found that the agricultural lands (PF and TF) and woodlands scatter across the lowland to uplands. Generally, the area percentage of PF and TF decreased with the increasing of elevation and slope, and that of WL has a significant opposite tendency (Fig S1).

In common, along with the comment of S. Rolinski, there are higher contents of soil nutrients in the lowlands than in the highlands. However, this is not true in our research area. We analyzed the changes of contents of soil SOC, TN and TP along the gradients of elevation and slope for the three types of land use of PF, TF and WL (Fig. S2-S3). For PF and TF, the contents of SOC, TN, and TP did not significantly change across the different gradients of elevation or slope. In contrast, the SOC and TN contents for WL in highlands at higher elevation and steeper slope were larger than under the same vegetation in the lowlands. Moreover, there were significant differences of SOC, TN and TP between the land uses PF, TF and WL within an individual class of elevation or slope (Fig. S2-S3). For a given elevation or slope group, the SOC, TN and TP contents at PF were generally larger than at the corresponding group at TF and WL, and TP contents were lowest at WL. Additionally, the TN contents for TF were lowest among three land use types, irrespectively of elevation or slope. This clearly indicates the impact of elevation and slope on nutrient stoichiometry in case of a near natural vegetation, while a transformation to agricultural use leads to characteristic changes of the stoichiometry, irrespectively of the site condition. We are therefore convinced that
our study is not biased due to the fact of preferential agricultural utilization of better soils at lower elevation.

We further analyzed the changes in C:N, C:P, and N:P molar ratios along the gradients of elevation and slope for the three types of land use of PF, TF and WL (Fig. S4-5). Except for WL, C:N, C:P, and N:P molar ratios of PF and TF did not significantly change across the different gradients of elevation or slope. But for WL, these stoichiometric ratios increased with increasing elevation and steepness of slope. Within individual classes of elevation or slope, the molar ratios of C:N and C:P between PF and TF had no significant differences, while the ones of C:P, and N:P for WL were always highest among the three land uses.

———————————————————

Figure S1. The area percent for paddy field (PF), tea farmlands (TF), and
woodland (WL) under different gradients of elevation and slope.

**Fig. 1.**

[Figure]

Figure S2. The contents of SOC, TN, and TP contents for paddy field (PF), tea farmlands (TF), and woodland (WL) under different elevation. Different uppercase letters denote significant differences among elevation classes for a given land use type. Different lowercase italic letters denote significant differences among land use types, within an individual elevation group (p < 0.05). Bars represent standard error.

**Fig. 2.**

[Figure]

Figure S3. The contents of SOC, TN, and TP for paddy field (PF), tea farmlands (TF), and woodland (WL) under different slope. Different uppercase letters denote significant differences among slope classes for a given land use type. Different lowercase italic letters denote significant differences among land use types, within an individual elevation (p < 0.05). Bars represent standard error.

**Fig. 3.**

[Figure]

Figure S4. The molar ratios of C:N, C:P, and N:P for paddy field (PF), tea farmlands (TF), and woodland (WL) under different elevation. Different uppercase letters denote significant differences among elevation classes for a given land use type. Different lowercase italic letters denote significant differences among land use types, within an individual elevation (p < 0.05). Bars represent standard error.

**Fig. 4.**

[Figure]

Figure S5. The molar ratios of C:N, C:P, and N:P for paddy field (PF), tea farmlands (TF), and woodland (WL) under different gradients of slope. Different uppercase letters denote significant differences among elevation classes for a given land use type. Different lowercase italic letters denote significant differences among land use types, within an individual elevation (p < 0.05). Bars represent standard error.

**Fig. 5.**